# Effects of Heat Treatment on Phase Formation in Cytocompatible Sulphate-Containing Tricalcium Phosphate Materials

**Dinara R. Khayrutdinova** [1],*[ID]**, Margarita A. Goldberg** [1],*[ID]**, Olga S. Antonova** [1]**, Polina A. Krokhicheva** [1]**, Alexander S. Fomin** [1][ID]**, Tatiana O. Obolkina** [1]**, Anatoliy A. Konovalov** [1]**, Suraya A. Akhmedova** [2]**, Irina K. Sviridova** [2]**, Valentina A. Kirsanova** [2]**, Natalia S. Sergeeva** [2]**, Sergey M. Barinov** [1] **and Vladimir S. Komlev** [1][ID]

[1] A.A. Baikov Institute of Metallurgy and Materials Science, Russian Academy of Sciences, 119334 Moscow, Russia

[2] P.A. Hertsen Moscow Oncology Research Institute—Branch of National Medical Research Radiological Centre affiliated with Ministry of Health of Russian Federation, 2nd Botkinsky Pr. 3, 125284 Moscow, Russia

*  Correspondence: dvdr@list.ru (D.R.K.); mgoldberg@imet.ac.ru (M.A.G.);
    Tel.: +7-(926)-406-3408 (D.R.K.); +7-(929)-651-6331 (M.A.G.)

**Abstract:** Powders based on β-tricalcium phosphate (β-TCP) containing sulphate groups at up to 12.0 mol.% were synthesised by chemical precipitation from aqueous solutions. The obtained materials were characterised by X-ray phase analysis, Fourier transform infrared spectroscopy, measurement of specific surface area, scanning electron microscopy, energy dispersive analysis, synchronous thermal analysis, mass spectra investigations and biological assays. It was established that during the synthesis, the obtained materials lose the sulphate groups in the course of heat treatment at 900 or 1200 °C. These groups stabilise low-temperature β-TCP, but when introduced at a high concentration, the sulphate groups contribute to the formation of hydroxyapatite during the heat treatment. Specific surface area of the powders proved to be in the range 81.7–96.5 $m^2/g$. Results of biological assays showed cytocompatibility of both pure β-TCP and samples of sulphate-containing β-TCP. Additionally, matrix properties in the culture of MG-63 cells were revealed in all samples. Thus, the obtained materials are promising for biomedical applications.

**Keywords:** wet chemical synthesis; thermal stability; β-tricalcium phosphate; hydroxyapatite; sulphate group; cytocompatibility



## 1. Introduction

Calcium phosphate is an attractive material for bone tissue substitution due to its chemical composition, which is similar to native bone tissue [1]. A promising synthetic material for dental and surgical applications is β-tricalcium phosphate $Ca_3(PO_4)_2$ (β-TCP) due to its biocompatibility, osteoconductive properties and a high resorption rate, which contribute to the replacement of an implanted scaffold on native new bone [2]. β-TCP is applied in the form of dense ceramic blocks [3], porous scaffolds [4], cement-forming materials [5] or a coating on metal implants [6,7] as a component of composite materials [8]. Introduction of various ions into β-TCP results in noticeable changes in the material's properties, which influence its biological behaviour [9,10]. For instance, introduction of magnesium cations leads to a significant increase in the strength of β-TCP cement [11], whereas the introduction of iron [12], copper [13], silver [14] or zinc [5] cations or their combinations [15] helps to enhance antibacterial properties. At the same time, modification of β-TCP with various anions also affects its structure and physicochemical and biological behaviour [16]. Fluorine and chlorine ions influence the sintering temperature of bioceramic materials and the resultant phase ratio of hydroxyapatite (HA) $Ca_5(PO_4)_3(OH)$ to

β-TCP [17]. Introduction of $SiO_4$ anions into a β-TCP coating for a stainless-steel implant increased the biomineralisation of the material along with HA phase formation [18]. As in the case of HA, a combination of β-TCP with calcium sulphate (CS) is of great research interest. The presence of sulphate groups can improve the solubility of β-TCP and thus the formation of composite cement based on CS and calcium phosphate for bone tissue replacement [19]. CS materials are characterised by a high resorption rate and biocompatibility, and are widely used as ceramic materials [20–22] and cement materials [23,24] as well as components of composites [25,26]. On the other hand, the addition of calcium phosphates to CS has a beneficial effect on its properties. For example, the addition of calcium nanophosphate at 1.0 wt.% increases the strength of cement two-fold and reduces its solubility [26]. In composite cement based on CS and HA at a phase ratio of 3:2, when an aqueous solution of chitosan is applied as a cement liquid, setting time of gypsum cements extends from 6 to 14 min [25]. To give another example, the decomposition of CS as a component of calcium phosphate microspheres accelerated the release of bone morphogenetic protein 2 (BMP-2), which enhanced the restoration of bone tissue in a defect area. At the same time, as a result of CS dissolution, a porous structure amounting to 34.3% ± 4.2% (mean ± SD) formed, and the pore size was 11.5 ± 2.4 μm [27]. Composite microspheres based on the β-TCP/CS system, and loaded with mixtures of vancomycin/tobramycin and gentamicin/tobramycin, possess antibacterial properties against *Pseudomonas aeruginosa* and *Staphylococcus aureus* for a period of up to 40 days [28]. The sulphate anion plays an important role in the development of inorganic phosphors, and holds promise for improving the luminescent properties of β-TCP powders [29]. Sulphate-substituted β-TCP has been synthesised by the wet precipitation method accompanied by microwave irradiation. The phase formation and powder morphology have been investigated, and the ability of β-TCP to incorporate $SO_4^{2-}$ has been demonstrated [30].

It should be noted that an important characteristic of all calcium phosphate materials is thermal stability, which makes it possible to predict their behaviour during a heat treatment, and properties of the sintered bodies. For example, during a heat treatment of natural HA, calcium oxide forms [31], which may reduce the viability of osteoblasts [32]. β-TCP materials are characterised by difficulties with sintering at high temperatures due to the allotropic phase transition from the rhombohedral β phase to monoclinic α-TCP starting from 1125 °C [33]. These problems lead to uncontrollable phase formation followed by cracking of the scaffolds because of thermal expansion [34]. Data on thermal behaviour of sulphate-substituted β-TCP are scarce, to the best of our knowledge. Studies have been performed only on calcium-deficient HA that is homogeneously resuspended at 2% or 3% concentration in a double volume of dilute aqueous ammonium sulphate and subjected to microwave treatment, with heat treatment temperatures of 900 and 1100 °C [35]. In the present work, β-TCP materials containing sulphate groups in a wide range of concentrations up to 12 mol.% were obtained by the wet co-precipitation technique, which allowed control of the chemical composition. Their physicochemical thermal stability and biological properties in vitro were investigated.

## 2. Materials and Methods

### 2.1. Powder Synthesis

The synthesis of sulphate-substituted β-TCP powders was carried out by the chemical precipitation method according to the following general equation:

$$3Ca(NO_3)_2 \cdot 4H_2O + 0.03x(NH_4)_2SO_4 + (2 - 0.02x)(NH_4)_2HPO_4 \rightarrow Ca_3(PO_4)_{(2-0.02x)}(SO_4)_{0.03x} \qquad (1)$$

where x is the content of sulphate groups (x = 0, 1.5, 3.5, 7.0, or 12.0).

The salts $Ca(NO_3)_2 \cdot 4H_2O$ and $(NH_4)_2HPO_4$ from Labtech (Russia), and $(NH_4)_2SO_4$ from Ruskhim (Russia), were used as starting reagents. Ammonium diphosphate and ammonium sulphate were dissolved in 50 mL of distilled water, and calcium nitrate was dissolved in 100 mL of distilled water. Then, the solutions of ammonium salts were slowly

added dropwise to the solution of nitrates at a constant pH of 6.9–7.4. The pH values were controlled by a Testo 206-pH1 (Germany).

Table 1 shows the assignment of samples and initial salt concentrations along with theoretical composition of the synthesised powders.

**Table 1.** Assignment of samples and synthesis specifics.

| Sample ID | Content of Sulphate Groups, mol.% (x) | Calculated Formula | $mCa(NO_3)_2 \cdot 4H_2O$, g | $m(NH_4)_2SO_4$, g | $m(NH_4)_2HPO_4$, g |
|---|---|---|---|---|---|
| CP1 | 0 | $Ca_3(PO_4)_2$ | 114.00 | - | 43.00 |
| CP2 | 1.5 | $Ca_3(PO_4)_{1.97}(SO_4)_{0.045}$ | 113.70 | 0.95 | 41.70 |
| CP3 | 3.5 | $Ca_3(PO_4)_{1.93}(SO4)_{0.105}$ | 112.90 | 2.20 | 40.60 |
| CP4 | 7.0 | $Ca_3(PO_4)_{1.86}(SO_4)_{0.21}$ | 111.70 | 4.40 | 38.70 |
| CP5 | 12.0 | $Ca_3(PO_4)_{1.76}(SO_4)_{0.36}$ | 110.00 | 7.40 | 36.01 |

After synthesis, the powders were vacuum-filtered in a Buchner funnel and dried at 300 °C until the liquid phase was completely removed. Additional heat treatments were carried out at 900 or 1200 °C at a heating rate of 6 °C/min with isothermal holding for 2 h in an air atmosphere.

## 2.2. Powder Characterisation

The amount of sulphur in the samples was determined by means of a Leco CS-600 gas analyser (Leco Corporation, St. Joseph, MO, USA) with an accuracy of 0.0001 mg. Iron chips (Leco cat. # 501-077) served as a catalyst. The materials were analysed after 300, 900 or 1200 °C heat treatment.

Specific surface area (SSA) was determined according to Brunauer, Emmet and Teller (BET) by low-temperature nitrogen adsorption measurements (Micrometric TriStar Analyzer, Micrometric Instruments, Norcross, GA, USA).

The powder materials were subjected to X-ray phase analysis (XRD) on a SHIMADZU-6000 diffractometer (Shimadzu Corporation, Kyoto, Japan) in the reflection mode (Bragg-Brentano geometry) using $CuK_\alpha$ radiation. Data from ICDD PDF-2 were used to determine the phase composition. Fourier transform infrared (FTIR) spectroscopy was performed on a Nicolet Avatar 330 FT-IR instrument (Thermo Fisher Scientific, Waltham, USA) in the range of 4000–400 cm$^{-1}$ in the diffuse reflection mode. The samples were prepared as mixtures with KBr.

Synchronous thermal analysis was performed on a Netzsch STA 409 PC Luxx (Netzsch, Selb, Germany) thermal analyser at a heating rate of 10 °C/min. Sample weight was at least 10 mg. Analysis of the composition of the gas phase resulting from the sample decomposition was carried out by means of a Netzsch QMS 403C Aëolos (Netzsch, Selb, Germany) quadrupole mass spectrometer coupled with a Netzsch STA 409 PC Luxx thermal analyser. Mass spectra were recorded for molecular weights 18 ($H_2O$) and 64 ($SO_2$).

Scanning electron microscopy (SEM) of the materials was performed using a Tescan VEGA II electron microscope (Tescan, Brno, Czech Republic) with an INCA Energy 300 energy dispersive analyser (Oxford Instruments Analytical, Abingdon, UK).

## 2.3. In Vitro Assays of Cytotoxicity and Cytocompatibility

To evaluate biological properties of the synthesised powder materials, in vitro assays were carried out to determine cytotoxicity in the presence of sample extracts (indirect contact method) and to evaluate cytocompatibility when cells were seeded on the samples (direct contact method) according to ISO 10993, part 5. Human osteosarcoma cell line MG-63 (Russian Collection of Cell Cultures, Institute of Cytology, Russian Academy of Sciences, St. Petersburg, Russia) was used for these purposes. For the in vitro assays, free-filled powder samples were heat treated at 1200 °C for 2 h. The resulting sinter was crushed in an agate mortar, and a fraction 470–1000 µm in size was separated with a nylon sieve. Next, the obtained particles were sterilised by γ-irradiation at a dose of 18 kGy.

### 2.3.1. The Indirect Contact Method

To prepare an extracting solution, 0.3 g of each type of sterile ceramic sample of β-TCP (CP1, CP2 or CP5) was placed in sterile vials, and 1.5 mL of the complete growth medium (CGM, consisting of DMEM (PanEco, Moscow, Russia), 10% of foetal calf serum (HyClone, Logan, UT, USA), 4 mM glutamine, a 1 M HEPES solution (PanEco, Moscow, Russia), and a gentamicin solution (50 μg/mL, Dalkhimpharm, Khabarovsk, Russia)) was added. The extraction was performed for 24 h at 37 °C with constant stirring on an orbital shaker (Elmi, Riga, Latvia). The seeding density of the test cells was $15.0 \times 10^3$ per well ($45.0 \times 10^3$ cells per cm$^2$, 96-well culture plates, Corning, NY, USA) in 200 μL of the CGM. At 24 h after cell seeding (i.e., after formation of a sub-confluent monolayer of MG-63 cells), the culture medium was removed from the wells, and 200 μL of each sample extract was added. CGM alone was added to the cells as a negative control, and a 20% dimethyl sulphoxide solution (DMSO, PanEco, Moscow, Russia) in the CGM was added as a positive control. For all experimental materials and controls, at least three duplicate tests were performed. The cells were grown in the presence of the extracts for 24 or 72 h.

### 2.3.2. The Direct Contact Method

Sterile samples of undoped β-TCP (CP1) and sulphate-substituted β-TCP (CP2 and CP5) were placed in 96-well culture plates (Costar, Washington, DC, USA). Four replicates were implemented for each group and period, including a control optical blank, and the cell suspension was added at a concentration of $20.0 \times 10^3$ per well (seeding density: $60.0 \times 10^3$ cells/cm$^2$). MG-63 cells were grown on the samples for 24, 72 or 168 h. Wells with cells on polystyrene culture plastic served as a control.

All procedures were performed under sterile conditions in an atmosphere of humid air at 37 °C and a flow of 5% $CO_2$ ($CO_2$ Sanyo Incubator, Tokyo, Japan). All experiments were conducted three times.

### 2.3.3. Cell Viability Assay

The 3-(4,5-dimethylthiazol-2-yl)-2,5-diphenyltetrazolium bromide (MTT) assay was performed to measure the viability of MG-63 cells. The details of this method have been described previously [36]. The 96-well culture plates with samples and cells after each incubation period were treated with MTT (5.0 mg/mL) (Sigma-Aldrich, Saint Louis, MO, USA) for 4 h at 37 °C. The absorbance of a formazan solution (reaction product) was determined on a microplate reader (Multiscan FC, Thermo Scientific, Waltham, MA, USA) at 540 nm. For each powder sample after a certain period of cell growth, the population of viable cells (PVC) was calculated relative to the control (in %) using the formula

$$PVC = OD_{exp}/OD_{control} \times 100\% \tag{2}$$

where $OD_{exp}$ and $OD_{control}$ are the optical densities of the formazan solution in the experiment and in the negative control, respectively. After that, according to ISO 10993-5:2009, the toxicity index (TI) was calculated as follows:

$$TI = 100\% - OD_{exp}/OD_{control} \; (\%) \tag{3}$$

A sample of a material was assumed to be non-toxic at a TI below 30% and cytocompatible at PVC $\geq$ 70%.

The results were processed by conventional methods of variation statistics in Microsoft Excel 2000. The significance of differences was assessed by Student's parametric $t$ test. Differences were considered statistically significant at $p < 0.05$.

## 3. Results

### 3.1. Powder Chemical Composition

The observed and calculated amounts of sulphur, along with SSA data, are given in Table 2. In terms of stoichiometry, each tested combination of reagents was found to react

completely. SSA was in the range of 81.7–96.5 m$^2$/g. Judging by the table, the SO$_4$ content did not affect SSA strongly.

**Table 2.** Theoretical and experimentally estimated amounts of sulphur involved in the synthesis of initial powders.

| Sample ID | Content of Sulphate Groups, mol.% | Amount of Sulphur (300 °C), wt.% | | SSA, m$^2$/g |
|---|---|---|---|---|
| | | Theoretical | Experimental * | |
| CP1 | 0.0 | 0.00 | 0.00 | 95.1 ± 0.1 |
| CP2 | 1.5 | 0.49 | 0.42 | 81.7 ± 0.1 |
| CP3 | 3.5 | 1.10 | 1.00 | 92.8 ± 0.1 |
| CP4 | 7.0 | 2.10 | 2.00 | 96.5 ± 0.1 |
| CP5 | 12.0 | 3.58 | 3.48 | 87.6 ± 0.1 |

* Uncertainty of 3%.

Table 3 shows experimental values of the sulphur amounts that remained after the heat treatment at 900 or 1200 °C. It was found that upon heating, the amount of sulphur noticeably diminished due to the thermal decomposition of the powders, as confirmed by the thermogravimetry (TG) data presented below. The significant loss of the sulphate amount was linked with the morphology of the materials, which were characterised by large SSA (Table 2).

**Table 3.** Amount of sulphur and the corresponding calculated amount of sulphate groups remaining after the heat treatment at 900 or 1200 °C.

| Sample ID | 900 °C | | 1200 °C | |
|---|---|---|---|---|
| | Measured Amount of Sulphur *, wt.% | Calculated Content of Sulphate Groups, mol.% | Measured Amount of Sulphur *, wt.% | Calculated Content of Sulphate Groups, mol.% |
| CP1 | 0.000 | 0.000 | 0.000 | 0.000 |
| CP2 | 0.019 | 0.057 | 0.005 | 0.015 |
| CP3 | 0.030 | 0.080 | 0.006 | 0.017 |
| CP4 | 1.200 | 3.600 | 0.188 | 0.560 |
| CP5 | 3.670 | 11.010 | 1.960 | 5.880 |

* Uncertainty of 3%.

## 3.2. XRD Analysis

According to XRD data after the synthesis, the only phase in CP1–CP4 materials was HA (ICDD PDF-2 No. 000-09-0432) (Figure 1). When 12 mol.% of sulphate groups was introduced (CP5), another phase appeared: CS hemihydrate (ICDD PDF-2 No. 000-01-0999). All the powders were characterised by low crystallinity.

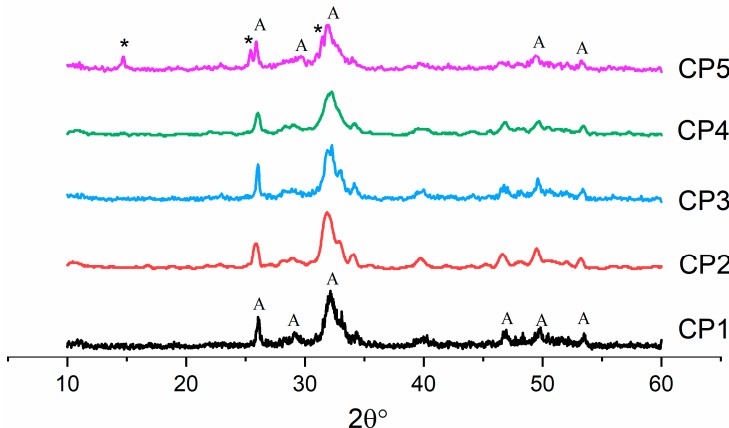

**Figure 1.** Diffractograms of the synthesis products (300 °C), where * is $CaSO_4 \cdot 0.5H_2O$, and A is $Ca_5(PO_4)_3(OH)$.

After the heat treatment at 900 °C, all powders contained β-TCP (ICDD PDF-2 No. 000-09-0169) as the main phase. In addition, all diffraction patterns pointed to the presence of a tiny amount (up to 3 wt.%) of calcium diphosphate $Ca_2P_2O_7$ (ICDD PDF-2 No. 000-09-0346), whose presence could be linked with the pyrolysis of a liquid in powders dried at 300 °C [37]. In ref. [38], it is shown that during wet chemical synthesis, the crystallinity of samples decreases, and a second phase, β-$Ca_2P_2O_7$, comes into being. For CP5, the formation of a third phase was observed, and β-TCP, $Ca_2P_2O_7$ and HA co-existed (Figure 2a). This phase's formation was facilitated by the presence of sulphate groups. A similar effect was documented in ref. [39], where researchers implemented mechanical mixing of initial components $CaHPO_4 \cdot 2H_2O$, $CaCO_3$ and $CaSO_4 \cdot 2H_2O$. $CaHPO_4 \cdot 2H_2O$ and $CaCO_3$ were employed in a stoichiometric ratio corresponding to TCP, and then gypsum ($CaSO_4 \cdot 2H_2O$) was added at 1.8–18.0 mol.%, and all reagents were mixed in an agate mortar [39]. Next, the resulting mixture was heated at 1300 °C. When the gypsum content was 7.2 mol.%, the emergence of HA was observed [39] as described by the reaction

$$3Ca_3(PO_4)_2 + CaSO_4 + H_2O \text{ (in air)} \rightarrow 2Ca_5(PO_4)_3(OH) + SO_3\uparrow \qquad (4)$$

In our work, during the heat treatment of pure β-TCP (CP1) at 1200 °C, there was a transition of β-TCP to α-TCP (ICDD PDF-2 No. 000-29-0359) (Figure 2b), as reported elsewhere [40]. When sulphate groups were introduced, the transition of β-TCP to α-TCP seemed to stop completely, and no peaks of α-TCP were detectable in CP2–CP5. Of note, the amount of $Ca_2P_2O_7$ decreased with the growth of the number of $SO_4$ groups. In ref. [35], researchers conducted a study about the influence of sulphate groups under microwave irradiation on the synthesis of β-TCP at degrees of substitution up to 3 mol.%. They also demonstrated a drop of the $Ca_2P_2O_7$ amount. This phenomenon was related to the reaction of β-TCP formation in the presence of $CaSO_4$:

$$2Ca_2P_2O_7 + 2CaSO_4 \rightarrow 2\beta\text{-}Ca_3(PO_4)_2 + 2SO_2\uparrow + O_2\uparrow \qquad (5)$$

In our work, the HA phase arose at a concentration of $SO_4^{2-} \geq 7$ mol.% for both CP4 and CP5 samples. The observed phase formation was confirmed by the FTIR data below.

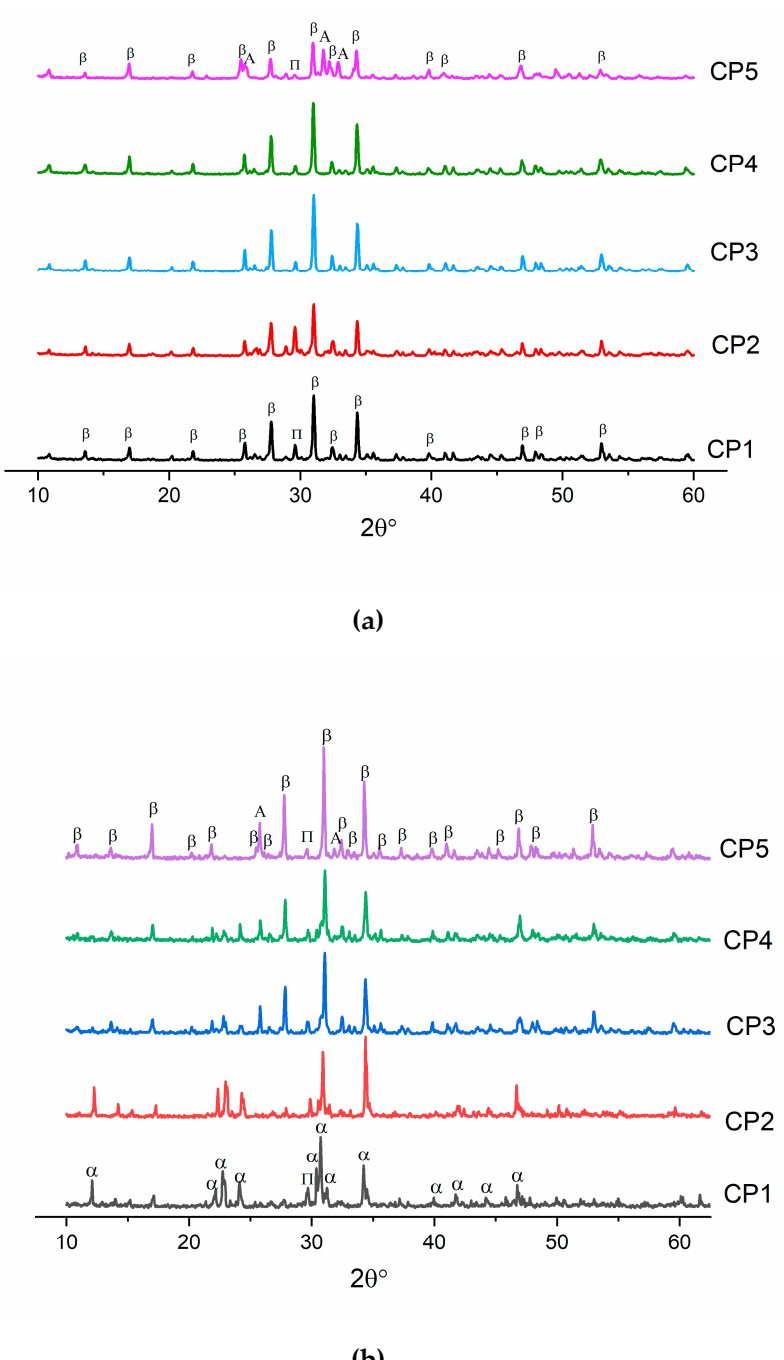

**Figure 2.** Diffractograms of samples heat-treated (**a**) at 900 °C or (**b**) at 1200 °C. β: β-TCP, α: α-TCP, A: hydroxyapatite, and Π: $Ca_2P_2O_7$.

Table 4 lists crystal lattice parameters of β-TCP powders heat-treated at 900 °C. One can see that the crystal lattice parameters diminish for samples CP2–CP3 compared to pure β-TCP, which may mean the incorporation of sulphate groups into β-TCP [39]. The decrease in lattice parameters was due to the introduction of a smaller sulphate ion (ionic radius 2.30 Å) in place of the phosphate ion (2.43 Å). A similar effect is described in ref. [29], where a substitution with 0.1 mol.% of $SO_4^{2-}$ also induced a decrease in lattice parameters, whereas at a higher degree of substitution (1.0 mol.%), the second phase ($CaSO_4$) arose. In our work, the substitution occurred at up to 0.08 mol.% of sulphate groups according to a decrease in lattice volume and the absence of additional $SO_4^{2-}$-containing phases. Past this point, there was a sharp expansion of volume, which was associated with a distortion

of the crystal lattice due to the formation of HA. By contrast, diffraction patterns of CP4 and CP5 were found to be shifted towards smaller angles, also indicating the emergence of the second phase.

**Table 4.** Crystal lattice parameters and crystallite size (D) of β-TCP powders heat-treated at 900 °C.

| Sample ID | a, nm | c, nm | V, nm$^3$ | D, nm |
|---|---|---|---|---|
| Theoretical * | 0.1043 | 0.3737 | 0.3527 | - |
| CP1 | 0.1043(3) | 0.3734(3) | 0.3521 | 43 |
| CP2 | 0.1043(1) | 0.3735(5) | 0.3519 | 43 |
| CP3 | 0.1042(3) | 0.3732(1) | 0.3511 | 33 |
| CP4 | 0.1043(3) | 0.3739(1) | 0.3524 | 38 |
| CP5 | 0.1045(1) | 0.3741(1) | 0.3538 | 44 |

* ICDD PDF-2 No. 000-09-0169.

The calculation of crystallite size (D) suggested that crystallites in the range of 33–44 nm were formed, and the introduction of the sulphate ion contributed to an increase in D insignificantly, consistent with the SSA data.

*3.3. FTIR Analysis*

According to FTIR spectroscopy of the synthesis products (300 °C), there were hydroxyl group bands at 3571 and 633 cm$^{-1}$, meaning the formation of HA [41] (Figure 3a). Bands of the $PO_4^{3-}$ group were also characteristic of the HA profile [42]. For instance, in the FTIR spectra, doublet $\nu_4$ was observed at 563 and 602 cm$^{-1}$, $\nu_3$ in the frequency range 1027–1111 cm$^{-1}$, and $\nu_1$ at 960 cm$^{-1}$. With an increase in the sulphate group content, the intensity of the peaks declined. The absorption region with a wave number of 870 cm$^{-1}$ belongs to $HPO_4^{2-}$. In addition to phosphate groups, there is a peak affiliated with the $NO_3$ groups related to by-products of the reaction: e.g., ammonium nitrate [43]. The peak at 1209 cm$^{-1}$ is typical for CO groups associated with heat treatment of synthesis products [44]. It is especially noteworthy that CP5 features a shoulder characteristic of $SO_4$ groups in the region of 670 cm$^{-1}$, thereby confirming XRD data of other researchers on the formation of the CS hemihydrate phase [45].

The FTIR spectra of the β-TCP powders calcined at 900 °C contained bands of the $PO_4^{3-}$ group, the profile of which is characteristic of β-TCP (Figure 3b). There was doublet $\nu_4$ at 550–610 cm$^{-1}$, mode $\nu_3$ was present in the frequency range 1040–1190 cm$^{-1}$, and $\nu_1$ at 946 and 972 cm$^{-1}$, in agreement with another report [46]. At the same time, CP5, which after the synthesis contained 12.0 mol.% of $SO_4^{2-}$, showed a profile of phosphate groups that became specific for HA, and the $\nu_3$ $PO_4^{3-}$ mode had a pronounced peak at 966 cm$^{-1}$. A band matching OH groups was observed at 3571 cm$^{-1}$. Aside from the phosphate groups, absorption bands belonging to sulphate groups at 649 and 674 cm$^{-1}$ were also registered in CP3, CP4 and CP5. The nature and behaviour of $SO_4$ bands are detailed in the Discussion section. For the powders of pure β-TCP, and at degrees of substitution of up to 7 mol.%, there were peaks of fluctuations of the P-O-P moiety at 727 and 1210 cm$^{-1}$, weakening with the growing content of sulphate groups. This phenomenon is due to the formation of β-TCP according to Equation (4) as a consequence of the interaction of calcium pyrophosphate with CS at >800 °C [47]. With an increase in the heat treatment temperature up to 1200 °C, materials CP4 and CP5 acquired split bands of $OH^-$ groups seen at 3563 cm$^{-1}$ (Figure 3c), which were suggestive of the incorporation of some sulphate groups into the HA lattice [48]. Additionally, in CP5, the shoulder at 633 cm$^{-1}$ is also associated with OH.

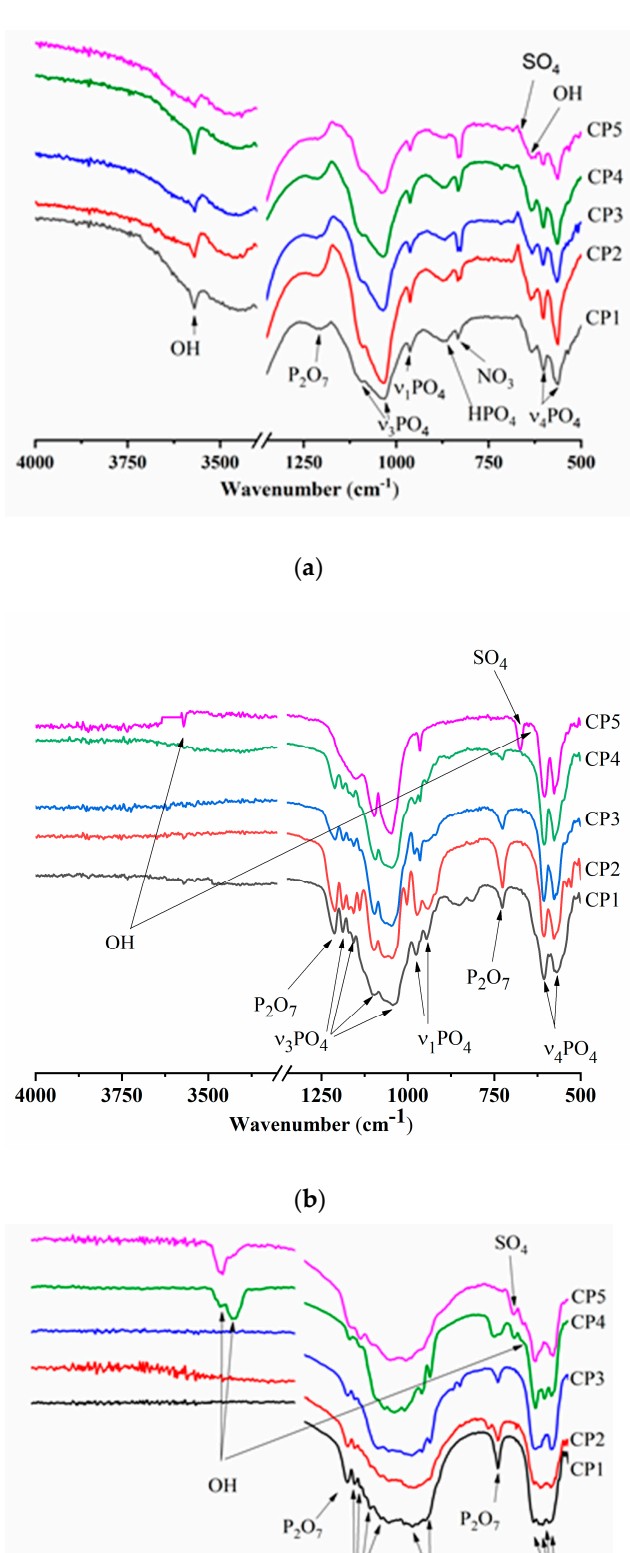

(**a**)

(**b**)

(**c**)

**Figure 3.** Infrared (IR) spectra of (**a**) synthesis products (300 °C), and (**b**) samples heat-treated at 900 °C or (**c**) at 1200 °C.

### 3.4. Synchronous Thermal Analysis and Mass Spectra

The behaviour of the mass loss curves up to 900 °C was similar among all samples except CP5, and was represented by three steps of mass changes: from 95 to 170 °C, from 350 to 490 °C and from 730 to 790 °C. According to the mass spectroscopy data, m/Z 18 ($H_2O$) corresponds to the loss of water, and Figure 4b illustrates the peaks with maxima at 170, 450 and 770 °C. The first peak matches the evaporation of physically bound adsorbed water, and the mass loss at this stage was 1.1–1.2 wt.% for samples CP1–CP4 and 2.3 wt.% for CP5. The next two stages involved gradual removal of crystalline water at higher temperatures. The last weight loss was caused by final removal of water and the transformation of calcium-deficient HA into tricalcium phosphate at ~770 °C [49]. The total weight loss at this stage was 5.0–5.2 wt.% for CP1–CP4 and 5.8 wt.% for CP5. Further increases in the temperature for the substituted materials led to the loss of the sulphate groups (Figure 4). The fourth step in the TG plots for CP2–CP5 corresponds to peaks with m/Z 64 ($SO_2$) with a maximum at 950–1000 °C. It should be noted that for CP3 and CP4 (a high content of sulphate groups), the release of $SO_2$ began at 760 °C, and for CP5, at an even lower temperature of 730 °C. The total weight losses were 5.2 wt.% for CP1, 6.1 wt.% for CP2 and CP3, 7.3 wt.% for CP4 and 10.6 wt.% for CP5.

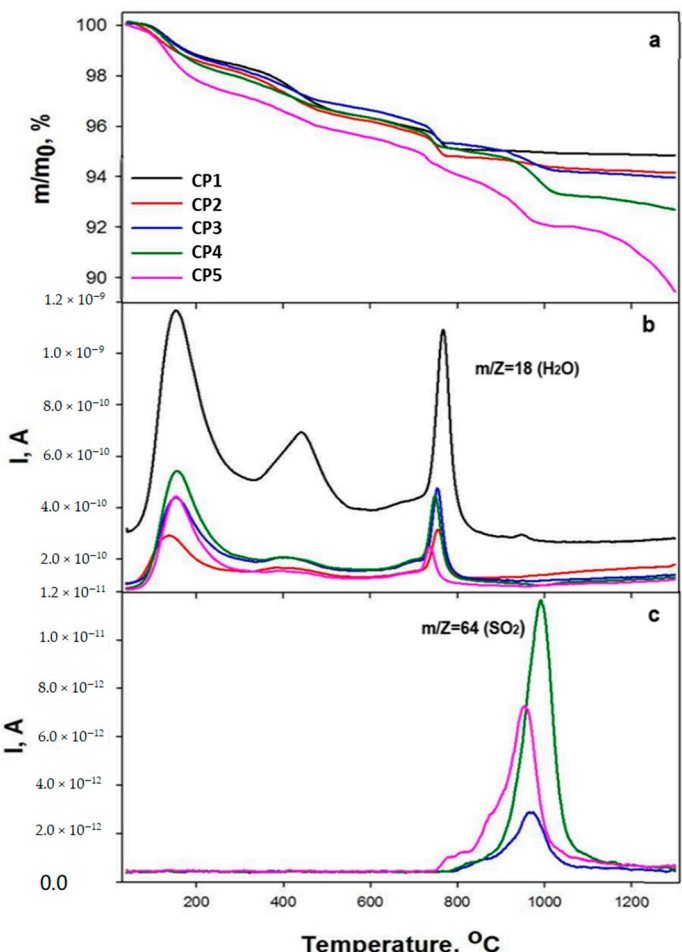

**Figure 4.** TG data (**a**), and mass spectra (**b**,**c**), of the powder samples.

It should also be mentioned that the temperature of the water losses differed between pure and substituted β-TCP. For the CP1 sample, the water losses started at a lower temperature and finished at a higher one, i.e., all peaks were wider compared to the peaks of the sulphate-doped samples. For instance, the first peak began at 67 °C for CP1 and at 82 °C for CP5, and the last peak ended at 826 °C for CP1 and at 777 °C for CP5. For all other samples, these points lay in between. Additionally, peaks of pure β-TCP had noticeably

higher intensity. It can be concluded that the content of sulphate anions contributes to a decrease in water crystallisation on the β-TCP surface. Nonetheless, because the mass loss during the evaporation of water was almost the same, this conclusion remains unconfirmed. Another explanation may be a release of NO (30) and $NO_2$ (46) from the initial reagents; these phenomena were revealed at 200–600 °C [50].

Our TG data confirmed the results of FTIR spectroscopy. For instance, for samples CP1–CP3 at 1200 °C, the peaks of the sulphate groups disappeared (Figure 3c). According to the TG data, the mass loss attenuation occurred at >1000 °C, whereas for samples CP4 and CP5, mass losses continued, which corresponded to the preservation of sulphate groups in the FTIR spectra. It is noteworthy that CP4 lost the sulphate group more slowly compared to CP5, as confirmed by the chemical analysis findings. This effect is due to the presence of CS in the CP5 synthesis product (Figure 1) that may decompose faster than $SO_4$-substituted TCP. For both CP4 and CP5 samples, mass losses continued even at temperatures above 1200 °C.

### 3.5. Morphological Analyses by SEM

The microstructure of the synthesis products above 300 °C for CP1 was characterised by a loose, flake-like morphology of particles with a size of <1 nm, which gave rise to agglomerates up to 1 μm (Figure 5a,b). According to energy dispersive analysis (EDA) data, the surface of the powder contained phosphorus, calcium and oxygen. The microstructure of CP5 was formed by two types of agglomerates and crystals: needle-like and flake-like (Figure 5d). EDA data from CP5 needle-like agglomerates showed the presence of Ca, S, O and P signals of low intensity, and agglomerates of the flake-like crystals were characterised by the co-existence of Ca, P, O and S, yielding signals of low intensity, indicating the formation of both sulphur- and phosphate-enriched phases after synthesis.

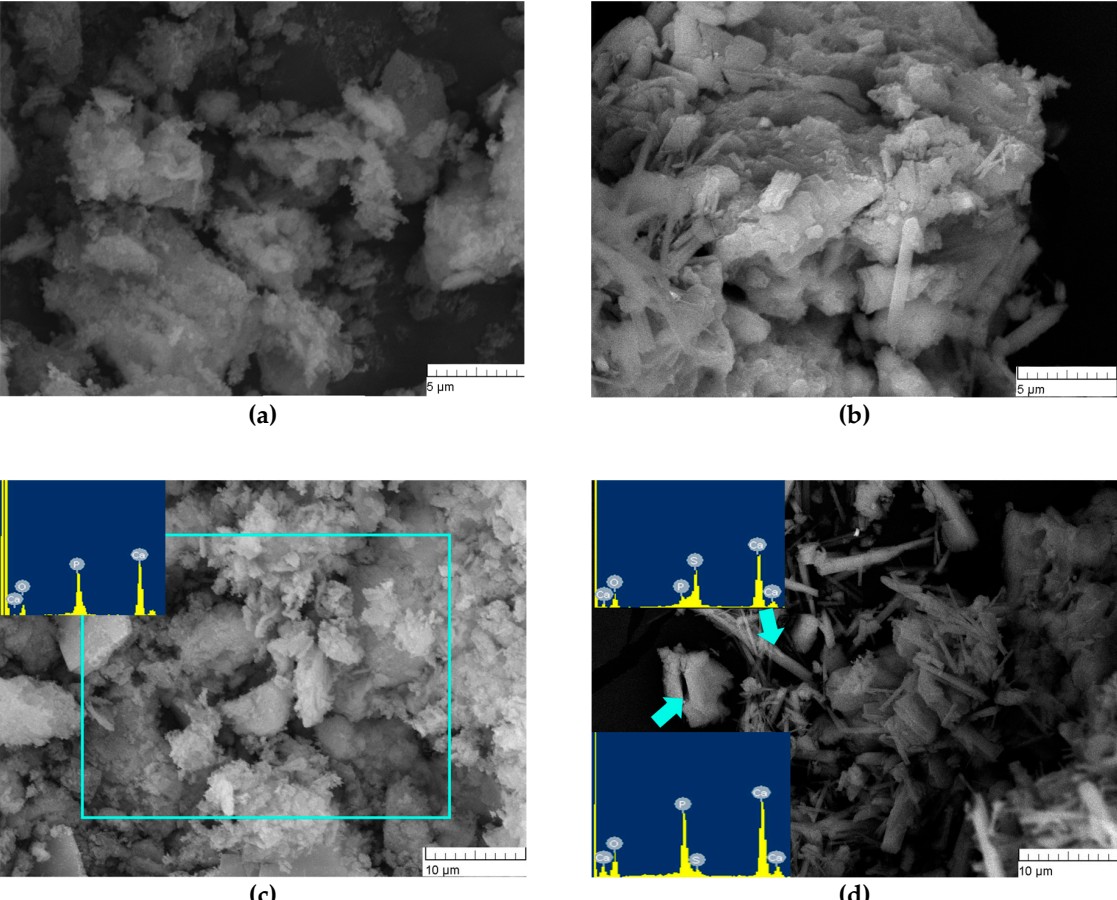

**Figure 5.** Microstructure of the synthesis products. (**a**,**c**): CP1, (**b**,**d**): CP5.

The morphology of the CP1 sample after the heat treatment at 900 °C featured agglomerates with a loose structure. EDA uncovered the presence of Ca, P and O at a Ca/P ratio of ~1.61 (Figure 6a) as in the case of 300 °C heat-treated products. The particles of CP5 were characterised by the presence of needle-like and flake-like crystals. The EDA findings suggested that the intensity of S and P peaks became similar for all examined zones, implying homogenisation of the chemical composition during the heating. Therefore, the powders consisted of elements Ca, P, S and O, in agreement with ref. [29], where powders were heat-treated at 1000 °C (Figure 6b).

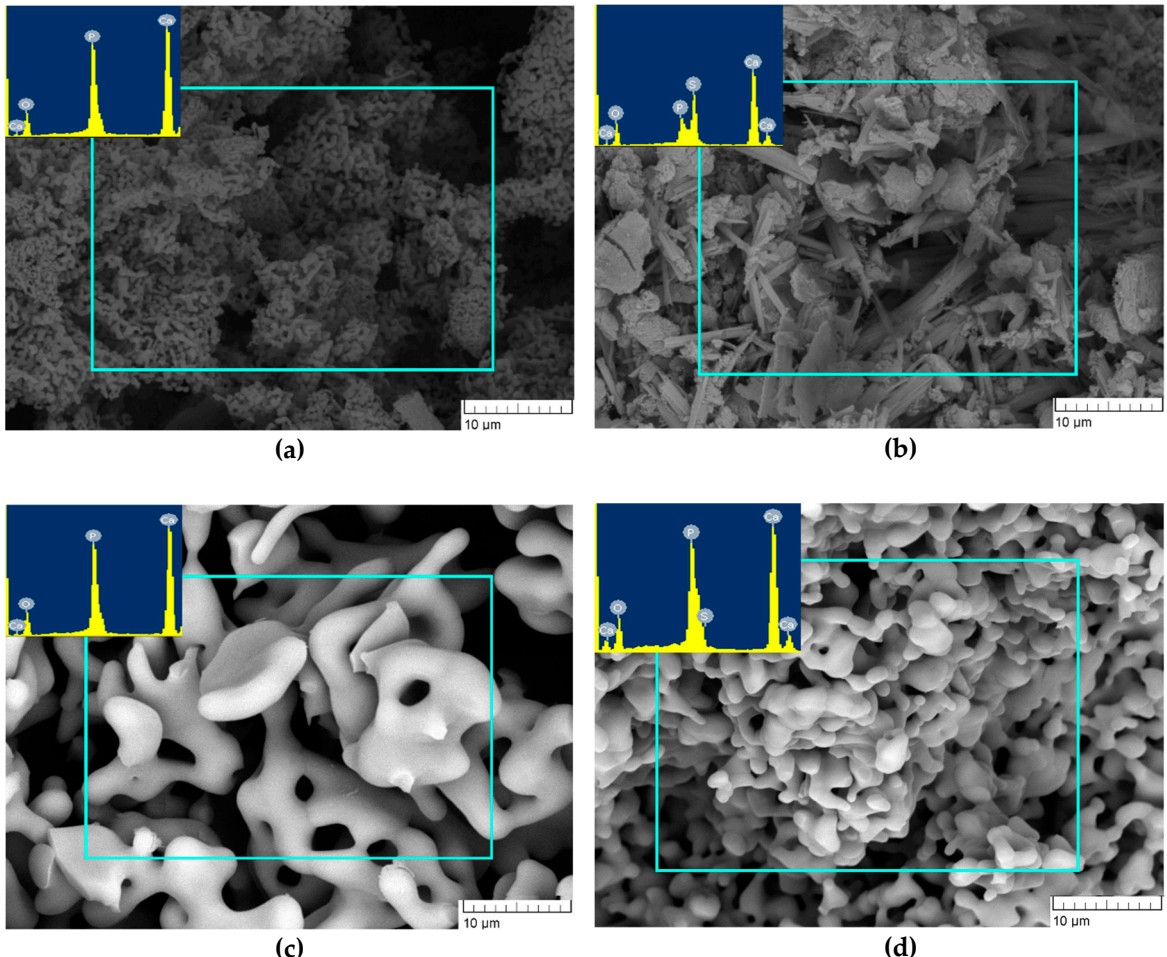

**Figure 6.** Morphology of powders heat-treated at 900 °C ((**a**): CP1, (**b**): CP5) or at 1200 °C ((**c**): CP1, (**d**): CP5).

After the heat treatment at 1200 °C, the powders of CP1 and CP5 featured the assembly of consolidated porous structure. The pore size in CP1 lay in the range of 2–5 μm, whereas CP5 pores had a size of up to 1 μm (Figure 6c,d). EDA of CP1 uncovered the presence of Ca, P and O, and the Ca/P ratio was 1.66, whereas CP5 manifested the presence of an S peak of very low intensity, suggesting that $SO_4$ persisted in CP5.

### 3.6. In Vitro Assays

It was shown that after 24 h of the growth of MG-63 cells in the presence of extracts of these materials, the pH of the samples was physiological, and the PVC and TI meant that the newly developed sulphate-containing materials were not toxic to the tested cultured cells (Table 5).

**Table 5.** pH values of extracts from the test samples, optical density of the formazan solution (OD, standard units, MTT assay), PVC, and the toxicity index (TI, %) towards the MG-63 test culture (24 h of incubation with the cells during their growth).

| Sample ID | pH | OD of Formazan Solution (Conventional Arbitrary Units) | PVC (%) | Toxicity Index (%) |
|---|---|---|---|---|
| Positive control-DMSO-20% | 7.4 | $0.062 \pm 0.02$ | 10.7 | 89.3 |
| Negative control-CGM | 7.4 | $0.578 \pm 0.01$ | 100.0 | 0.0 |
| CP1 | 7.8 | $0.457 \pm 0.01$ | 79.1 | 20.9 |
| CP3 | 7.5 | $0.528 \pm 0.02$ | 91.3 | 8.7 |
| CP5 | 7.5 | $0.485 \pm 0.03$ | 83.9 | 16.1 |

OD data are presented as mean $\pm$ SD.

For the in vitro assay, the powders were sintered at 1200 °C as in the case of pure β-TCP and cation-doped β-TCP materials [51]. According to our SEM data, the sintering at 1200 °C ensured the formation of a consolidated body that could serve as a porous scaffold, and this sintering did not lead to artifact cytotoxicity because of the presence of the materials in the form of powder particles [52].

Figure 7 shows histograms of the samples for different time points of MG-63 osteosarcoma cell cultivation, suggesting that the fastest cell growth was observed on CP3 for all cultivation periods. This finding is due to the lower solubility of CP3 compared to CP5, which retained CS. It is known that the presence of CS enhances the solubility of β-TCP [53]. Overall, the results of our experiments in vitro indicate the cytocompatibility of both pure β-TCP and samples of β-TCP containing sulphate groups. On the other hand, more pronounced matrix properties in relation to MG-63 cell culture were revealed in the CP3 sample. The high solubility of β-TCP is detrimental to cells due to a sharp release of calcium ions into the medium [54] (Table 5).

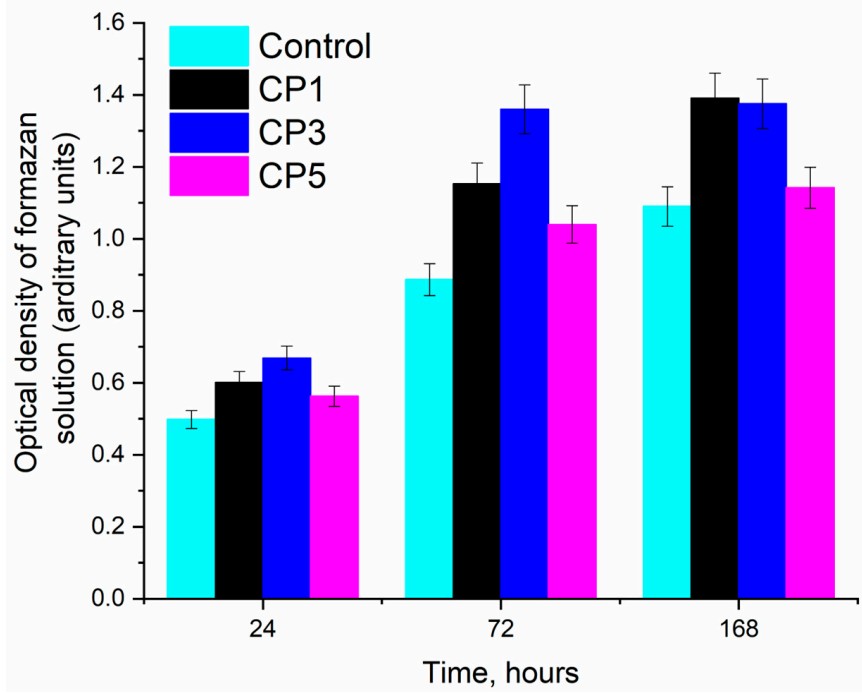

**Figure 7.** Optical density of the formazan solution (MTT assay, a.u.) and (in % relative to control) during the cultivation of human osteosarcoma MG-63 cells under control conditions and with the samples during the cultivation.

## 4. Discussion

Many studies have been devoted to the influence of doping of β-TCP with various ions on its physical, chemical, biological and other properties. The most popular have included the introduction of metal cations from various sources [15,55,56], while anionic doping of β-TCP has been the subject of only a limited number of articles [29,57,58]. Similar studies have been conducted mainly to assess the impact of sulphate groups on HA properties [46,58,59].

Our results suggest that wet chemical synthesis of β-TCP powders makes it possible to obtain one-phase powders with 7.0 mol.% of $SO_4$. The heating of the materials after synthesis up to 900 °C resulted in preservation of the sulphate groups, but according to XRD data, $SO_4$ groups diminished crystalline cell volume only when added at ≤0.08 mol.%. These data are in agreement with the results reported about materials obtained by mechanochemical synthesis, which allows for introduction of no more than 1.0 mol.% of this substituent [29,60]. Furthermore, the wet precipitation technique applied here yielded highly dispersed powders with SSA up to 96.5 $m^2$/g (for samples heat-treated at 300 °C).

For the first time (to the best of our knowledge), XRD, FTIR and TG analyses and mass spectroscopy, as well as chemical composition assays after the heat treatment of nanosized powders, were used to assess effects and behaviour of sulphate groups in β-TCP during heating for a wide range of composition of the initial mixture of reagents. Previously, TG analyses have been performed on an HA-$CaSO_4$ composite synthesised at pH 10 [47], and suggested that the main mass losses take place in the temperature range 80–300 °C during the decomposition of $CaSO_4·2H_2O$ to $CaSO_4$. In ref. [61], TG data are reported for a mixture of 1 mol.% of HA and 3 mol.% of $(NH_4)_2SO_4$ heated at 1100 °C in a helium atmosphere, where it was shown that the mass loss proceeds in four stages; this finding is comparable with our results.

Although FTIR spectroscopy usually enables investigators to clarify whether substitutions occur in the lattice of calcium phosphates, in the case of sulphate, there are some complications because bands of sulphate groups in IR spectra overlap substantially with bands of phosphate groups, and the high intensity of phosphate vibrations does not allow extraction of sulphate vibrations reliably. A possible exception is the vibration bands of sulphate groups in the region of 674 and 640 $cm^{-1}$, which characterise their out-of-plane vibrations. Such bands are characteristic of gypsum and other CSs [62], and their emergence may be explained by the formation of a separate phase: CS hemihydrate. In the case of substitutions in β-TCP or HA, the authors of ref. [29] could not detect these bands. Apparently, when sulphate groups were introduced into the lattice of calcium phosphates, the out-of-plane vibrations of sulphate almost disappeared. We confirmed the presence of sulphur up to 1200 °C in our samples by a high-precision method of elemental analysis. Furthermore, the retention of sulphur in the powders was indirectly evidenced by the incompleteness of the mass loss seen in our TG curves. Another possible explanation of the sulphur presence comes from other studies. Radha et al. carried out Raman analyses, which helped to detect isolated vibrations of sulphate groups (an S–O peak at 1008 $cm^{-1}$), whose intensity correlated with the sulphur content [63]. The authors of ref. [29] measured the second harmonic generation signal to investigate the vacancy at the M4 site of β-TCP and demonstrated its changes depending on $SO_4$ content.

In clinical practice, a commercial composite material is applied consisting of a mixture of CS and β-TCP: PRO-DENSETM (Wright Medical Technology, Memphis, TN, USA). Studies [64,65] showed complete healing of bone tissue in a defect zone. In our in vitro assays, the β-TCP materials containing sulphate groups were evaluated and found to have good properties, suggesting possible application of the obtained powder materials in medicine as promising biomaterials for the treatment of bone tissue defects.

## 5. Conclusions

By the co-precipitation technique, powder materials were obtained having large SSA (up to 96.5 $m^2$/g) and based on β-TCP with $SO_4^{2-}$ groups introduced at 1.0, 3.5,

7.0 or 12.0 mol.%. The product of the synthesis had an apatite structure for materials containing up to 7.0 mol.% of $SO_4^{2-}$, whereas an increase in the sulphate anion content up to 12.0 mol.% caused precipitation of CS as a second phase. The introduction of sulphate groups affected the microstructure and morphology of particles of the powder materials after synthesis, and the behaviour of these groups was influenced by heat treatment in the range of 900–1200 °C. It was found that the introduction of sulphate groups at up to 3.5 mol.% during the synthesis resulted in β-TCP phase stabilisation, whereas with an increase in their concentration up to 12.0 mol.%, a second phase (HA) arose during the heat treatment. For the first time (to the best of our knowledge), XRD, FTIR and TG analyses, mass spectroscopy, and chemical composition assays after a heat treatment of the nanosized powders showed the behaviour of the sulphate groups in β-TCP during heating. Sulphate evaporation started at lower temperatures when a large amount of sulphate was introduced. At the same time, the heating did not lead to a complete loss of the sulphate. The decomposition of β-TCP did not give rise to toxic phases such as CaO. From the results of in vitro experiments, it can be concluded that the newly developed powder materials are non-toxic and are fully cytocompatible.

**Author Contributions:** Conceptualization, D.R.K. and M.A.G.; methodology, O.S.A., P.A.K. and T.O.O.; validation, D.R.K. and N.S.S.; investigation, S.A.A., I.K.S., V.A.K., A.A.K. and A.S.F.; data curation, D.R.K. and A.S.F.; writing—original draft preparation, D.R.K., O.S.A., P.A.K. and T.O.O.; writing—review and editing, M.A.G., S.M.B. and V.S.K. All authors have read and agreed to the published version of the manuscript.

**Funding:** This research was funded by the Russian Science Foundation, grant number 22-79-10293.

**Data Availability Statement:** Data sharing not applicable.

**Acknowledgments:** The authors are grateful to Tatiana Borisovna Shatalova for the synchronous thermal analysis and the investigation of the mass spectra.

**Conflicts of Interest:** The authors declare no conflict of interest.

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
