# Peer review of "Effects of Heat Treatment on Phase Formation in Cytocompatible Sulphate-Containing Tricalcium Phosphate Materials"

_minerals, doi:10.3390/min13020147_

Round 1

Reviewer 1 Report

The paper deals with an interesting subject, which is the synthesis of powder based on β-tricalcium phosphate (TCP) containing sulfate groups.

The authors performed a deep phisico-chemical and morphological characterization by X-ray, FTIR, SEM-EDS, synchronous thermal and massa spectra analysis. Furthermore, in vitro tests were performed by using MG63 through direct and indirect tests. The paper is of clear interest for the readers of the journal as well as all the biomaterials community.

Nevertheless, the paper is not suitable for publication in the present form and some minor changes should be properly addressed by the authors to improve the clarity and the understanding.

1.  The authors should improve the English, in order to improve the readability of the manuscript.

2.     The authors should cite the acronyms the first time they appear in the main text.

3.   The authors should improve the abstract, highlighting some of their achievements rather than report the performed tests.

4.     The authors should improve the “Introduction” section. In particular, it results confusing and does not flow properly. The authors should report other papers, comparing the results, and highlighting the novelty of their work.

5.  The authors should improve the quality of Figure 3, 4, 6 and 7. Specifically, the images should be aligned.

6.     I would suggest also to reorganize the figures 3 and 4 in a panel. In such a way, the final number of figures would be reduced with the consequence to have a more clear and nice paper.

7.     With regard to the biological test, in the indirect method how did the authors evaluate the citocompatibility? Did they use MTT assay? Methods are missing. Accordingly, the results should be more discussed. Specifically the authors only reported table 5, in which the label of “OD solution formazan…” is not clear.

8.   Similarly, for direct tests, the authors reported in the methods a “population of viable cells” (PVC) calculation. However, the results are expressed as OD values. The authors should homogenize the information. Furthermore, the time should be homogenized (1, 3 and 7 days) instead of (24, 72, 168 h).     

Author Response

We would like to thanks the anonymous reviewer t the valuable comments.

  1. The authors should improve the English, in order to improve the readability of the manuscript,

Corrected

  1. The authors should cite the acronyms the first time they appear in the main text.

Corrected

  1. The authors should improve the abstract, highlighting some of their achievements rather than report the performed tests.

The authors added the main results to the abstract:

The SSA powders were in the range of 81.7-96.5 m2/g. The results of biological tests showed the cytocompatibility of both pure β-TCP and sulfate-containing β-TCP samples. At the same time, the matrix properties for the MG63 culture were revealed in all samples. These results showed that the obtained materials are promising for biomedical applications.

  1. The authors should improve the “Introduction” section. In particular, it results confusing and does not flow properly. The authors should report other papers, comparing the results, and highlighting the novelty of their work.

Expanded the introduction by adding information on the effect of calcium phosphates on calcium sulfate

  1. The authors should improve the quality of Figure 3, 4, 6 and 7. Specifically, the images should be aligned.

Corrected

  1. I would suggest also to reorganize the figures 3 and 4 in a panel. In such a way, the final number of figures would be reduced with the consequence to have a more clear and nice paper.

We reorganized the figures 3 and 4 into one figures â„– 3

  1. With regard to the biological test, in the indirect method how did the authors evaluate the citocompatibility? Did they use MTT assay? Methods are missing. Accordingly, the results should be more discussed. Specifically the authors only reported table 5, in which the label of “OD solution formazan…” is not clear.

MTT test was performed.

Added paragraph 2.3.3, which describes the methodology of the MTT test:

The MTT assay was performed to measure the viability of the MG-63 cell line. The details of this method were described previously [36]. Plates with samples and cells at each incubation period were treated with MTT (5.0 mg/ml) (3-(4,5-dimethylthiazol-2-yl)-2,5-diphenyltetrazolium bromide) (Sigma-Aldrich, USA) and then incubated for 4 hours at 37 °C. The absorbance of formazan solution (reaction product) was determined using a microplate reader (Multiscan FC, Thermo Scientific, USA) at 540 nm.

  1. Similarly, for direct tests, the authors reported in the methods a “population of viable cells” (PVC) calculation. However, the results are expressed as OD values. The authors should homogenize the information. Furthermore, the time should be homogenized (1, 3 and 7 days) instead of (24, 72, 168 h).

In table 5, add a column with the results of PVC (%). The authors corrected the information about MTT-test: 24, 72, 168 h.

Reviewer 2 Report

In my humble opinion, this well-written manuscript might be published almost as is. Just minor improvements are necessary:

1. The manuscript is devoted to sulfate-containing tricalcium phosphate, while the initial part of Introduction describes many irrelevant cases of ion-substituted calcium phosphates with references to the previously published papers written by the same authors. This is a hidden way of self-citation. Please, re-write the first part of Introduction by replacing improper initial 18 citations by additional amount of those on calcium sulphate, calcium sulphate/calcium phosphate mixtures or composites and phosphate/sulphate ionic substitutions.

2. Line 163: “Sig-ma-Aldrich” – please, correct

3. Fig. 6: SEM pictures are of a low quality, a and d especially. Similar is valid for Fig. 7a.

4. These are problems with subscripts in chemical formulas mentioned within the list of references. Namely:

29. D. V. Deyneko et al., “«Ellestadite»-type anionic [PO4]3– → [SO4]2– substitutions in β-Ca3(PO4)2 type compounds ….

64. D. J. Hall, T. M. Turner, and R. M. Urban, “Healing bone lesion defects using injectable CaSO4/CaPO4-TCP bone graft substitute ….

Please, re-check carefully the entire list of references.

Author Response

We are grateful to anonymus reviwer to the valuable comments.

  1. The manuscript is devoted to sulfate-containing tricalcium phosphate, while the initial part of Introduction describes many irrelevant cases of ion-substituted calcium phosphates with references to the previously published papers written by the same authors. This is a hidden way of self-citation. Please, re-write the first part of Introduction by replacing improper initial 18 citations by additional amount of those on calcium sulphate, calcium sulphate/calcium phosphate mixtures or composites and phosphate/sulphate ionic substitutions.

Replaced of self-citations in first 18 citations.

Expanded the introduction by adding information about calcium sulfate-calcium phosphate composits:

At the same time, the addition of calcium phosphates to calcium sulfate has a beneficial effect on its properties. Thus, the addition of calcium micronanophosphate in the amount of 1 wt. % increases the strength of SC cements by 2 times, and also reduces their solubility [26]. In a composite cement based on SC/HA at a phase ratio of 3:2, respectively, and using an aqueous solution of chitosan as a cement liquid, the setting time of gypsum cements increased from 6 to 14 min [25].

  1. Line 163: “Sig-ma-Aldrich” – please, correct,

Corrected

  1. 6: SEM pictures are of a low quality, a and d especially. Similar is valid for Fig. 7a.

Corrected

  1. These are problems with subscripts in chemical formulas mentioned within the list of references. Namely:
  2. D. V. Deyneko et al., “«Ellestadite»-type anionic [PO4]3– → [SO4]2– substitutions in β-Ca3(PO4)2 type compounds ….
  3. D. J. Hall, T. M. Turner, and R. M. Urban, “Healing bone lesion defects using injectable CaSO4/CaPO4-TCP bone graft substitute ….

Please, re-check carefully the entire list of references,

Corrected

Reviewer 3 Report

Dear Authors,

interesting work on tri-calcium phosphate, with a multi-methodological characterization based on more techniques, e.g. XRD FTIR SEM and so on, complete also of biological tests. According to my experience, I accept it with minor revisions (see below). Regards

L19: Fourier Transform Infrared spectroscopy

L25: SSA stands for ? Please do not use the acronym here

L35: please quote the chemical formula of TCP

L36: I would quote the similarity of TCP density with those of bone and teeth

L38:  ceramic blocks [3]  

L39: on the metal implants [6-7]

L 42: resulted in a significant increase of the strength of the β-TCP cement [11]

L44:  silver[14], and zinc [5] cations, as well as its combinations [15], contributed in the increasing of antibacterial properties 

L47: please quote the chemical formula of hydroxyapatite

L50; apatite or hydroxyapatite? If, as I think, it is the second, use the acronym HA.

L57: micro or nano? I thing is better nano phosphate

58: increases the strength of the cement 2 times ... as for lie 42, I would say 'significant incerase'

L95 equation 1: commas should be replaced by points for the decimals

L105: please fix the first row of Table 1, especially 'The content of sulfate groups ...'

L121: Fourier Transform Infrared (FTIR) spectroscopy

L181: if you quote Excel, I t should be better to quote also its reference.

L194: as for L105, same problem of formatting for first row. Please try to fix it.

L203: also in this case, please fix the first line of the Table.

L206-210. Quite confusing. 'the only phase for CP1-CP4 materials was poor crystalline HA' what does mean? 

Maybe 'In CP1-CP4 samples there is the presence of HA peaks? And how do you measure the 'crystallinity' of this HA? Just fro the sharpness of the peaks?

L208: ... of the emihydrate calcium sulfate 

L209: Again, how do you estimate this 'low degree of crystallinity' ?

L211: Figure 1 caption. Theta, use the greek symbol. Degree, use °.

L216: calcium diphosphate 

L218: again, please comment this lowering of crystallinity 

L237: hydroxyapatite ?

L240: Please enlarge the two figures. Too little, they should be placed 'in column'.

In the caption, 'diffraction patterns'

L246: ionic radius

L256: Write 'theoretical' and move 'ICDD PDF-2 ...' in the caption below with an asterisk.

L257. Crystalline size. How did you estimate the crystalline size ?

L291: Please, enlarge these FTIR figures. They should stay in column too. after the enlargement.

L292: IR - spectra of:  a) synthesis products (300 °C); b) samples heat-treated at 900°Ð¡ and. c), at 1200°Ð¡. 

L338. Please fix EDA/EDX acronyms here.

L354. Please report a table with EDX data.

L355. Morphology of powders heat-treated on 900 °Ð¡:  Ð°) CP1; b) CP5; and on 1200°Ð¡: c) CP1, (d) CP5.

L438: which suggested the possible application of the obtained powder materials in medicine as a promising biomaterial for the treatment of bone tissue defects.

(I suggest to move this sentence in the Conclusions, at the end, or to simply reply it a line 459) 

Author Response

Dear Authors,

interesting work on tri-calcium phosphate, with a multi-methodological characterization based on more techniques, e.g. XRD FTIR SEM and so on, complete also of biological tests. According to my experience, I accept it with minor revisions (see below). Regards

L19: Fourier Transform Infrared spectroscopy

Corrected

L25: SSA stands for ? Please do not use the acronym here

Corrected

L35: please quote the chemical formula of TCP

Corrected

L36: I would quote the similarity of TCP density with those of bone and teeth

Thank you for the interesting suggestion, we will consider it for the future. In this work, reference [2] indicates the physicochemical characteristics of TCP. At the same time, from many literature data, it is known that bone density and TCP are close.

L38:  ceramic blocks [3]  

Corrected

L39: on the metal implants [6-7]

Corrected

L 42: resulted in a significant increase of the strength of the β-TCP cement [11]

Corrected

L44:  silver[14], and zinc [5] cations, as well as its combinations [15], contributed in the increasing of antibacterial properties 

Corrected

L47: please quote the chemical formula of hydroxyapatite

Corrected

L50; apatite or hydroxyapatite? If, as I think, it is the second, use the acronym HA.

Corrected

L57: micro or nano? I thing is better nano phosphate

Corrected

58: increases the strength of the cement 2 times ... as for lie 42, I would say 'significant incerase'

Corrected

L95 equation 1: commas should be replaced by points for the decimals

Corrected

L105: please fix the first row of Table 1, especially 'The content of sulfate groups ...'

Corrected

L121: Fourier Transform Infrared (FTIR) spectroscopy

Corrected

L181: if you quote Excel, I t should be better to quote also its reference.

We do not quote Excel

L194: as for L105, same problem of formatting for first row. Please try to fix it.

Corrected

L203: also in this case, please fix the first line of the Table.

Corrected

L206-210. Quite confusing. 'the only phase for CP1-CP4 materials was poor crystalline HA' what does mean? 

Maybe 'In CP1-CP4 samples there is the presence of HA peaks? And how do you measure the 'crystallinity' of this HA? Just fro the sharpness of the peaks?

Corrected

L208: ... of the emihydrate calcium sulfate 

Corrected

L209: Again, how do you estimate this 'low degree of crystallinity' ?

We estimate this 'low degree of crystallinity' just fro the sharpness of the peaks.

L211: Figure 1 caption. Theta, use the greek symbol. Degree, use °.

Corrected

L216: calcium diphosphate 

Corrected

L218: again, please comment this lowering of crystallinity 

We estimate this 'low degree of crystallinity' just fro the sharpness of the peaks.

L237: hydroxyapatite ?

Corrected

L240: Please enlarge the two figures. Too little, they should be placed 'in column'.

In the caption, 'diffraction patterns'

Corrected

L246: ionic radius

Corrected

L256: Write 'theoretical' and move 'ICDD PDF-2 ...' in the caption below with an asterisk.

Corrected

L257. Crystalline size. How did you estimate the crystalline size ?

We calculated the size of the coherent scattering region from the diffractogram using the Scherrer formula.

L291: Please, enlarge these FTIR figures. They should stay in column too. after the enlargement.

Corrected

L292: IR - spectra of:  a) synthesis products (300 °C); b) samples heat-treated at 900°Ð¡ and. c), at 1200°Ð¡. 

Corrected

L338. Please fix EDA/EDX acronyms here.

Corrected

L354. Please report a table with EDX data.

Thanks for the suggestion, we'll keep that in mind for the future. But our task was to show exactly the elemental composition of the material by the EDA method, which was achieved.

L355. Morphology of powders heat-treated on 900 °Ð¡:  Ð°) CP1; b) CP5; and on 1200°Ð¡: c) CP1, (d) CP5.

Corrected

L438: which suggested the possible application of the obtained powder materials in medicine as a promising biomaterial for the treatment of bone tissue defects.

(I suggest to move this sentence in the Conclusions, at the end, or to simply reply it a line 459) 

Corrected

Round 2

Reviewer 1 Report

The authors clearly addressed all the questions from the reviewer. The paper can be accepted in the present form.

Reviewer 2 Report

A revised version of the manuscript looks better and, unless other reviewers find additional imperfections, in my humble opinion, this version might be published as is. No further corrections are necessary.

Reviewer 3 Report

Dear authors,

you followed almost all my advices. Just you could spend some extra word on size crystallite calculation, as I requested in the revision, but it is ok.

Regards Reviewer